# Analysis of Genetic Diversity and Resistance to Foliar Pathogens Based on Genotyping-by-Sequencing of a Para Rubber Diversity Panel and Progeny of an Interspecific Cross

**DOI:** 10.3390/plants11243418

**Published:** 2022-12-07

**Authors:** C. Bindu Roy, Shashi N. Goonetilleke, Limiya Joseph, Anu Krishnan, Thakurdas Saha, Andrzej Kilian, Diane E. Mather

**Affiliations:** 1Rubber Research Institute of India, Kottayam 686 009, India; 2School of Agriculture, Food and Wine, Waite Research Institute, The University of Adelaide, Glen Osmond, SA 5064, Australia; 3Diversity Arrays Technology, Canberra, ACT 2617, Australia

**Keywords:** disease resistance, genetic map, *Hevea brasiliensis*, KASP assay, marker-trait associations, phylogeny, potentially key disease resistance genes

## Abstract

Para rubber trees (*Hevea brasiliensis*) are the largest major source of natural rubber in the world. Its major pathogens are *Phytophthora* spp., *Corynespora cassiicola*, and *Colletotrichum* spp. A rubber diversity panel of 116 clones using over 12,000 single nucleotide polymorphisms (SNPs) from DArTSeq genotyping revealed clear phylogenetic differences in clones that originated from different geographical regions of the world. An integrated linkage map constructed with an F_1_ progeny of 86 from an interspecific cross between *H. brasiliensis* and *H. benthamiana* using 23,978 markers [10,323 SNPs and 13,655 SilicoDArTs] spanned 3947.83 cM with 0.83 cM average marker-interval. The genome scaffolds that were anchored to the linkage map, covering 1.44 Gb of *H. brasiliensis* reference genome, revealed a high level of collinearity between the genetic map and reference genome. Association analysis identified 12 SNPs significantly associated with the resistance against *Phytophthora*, *Corynespora,* and *Colletotrichum* in six linkage groups: 2, 6, 12, 14, 17, and 18. Kompetitive Allele-Specific PCR marker assays were developed for those 12 SNPs, screened with 178 individuals, and detected clear separation between two genotypes. Within the proximity to those SNPs, 41 potentially key genes that have previously been reported to associate with plant disease resistance were predicted with high confidence.

## 1. Introduction

*Hevea brasiliensis*, the Para rubber tree, is the only source of latex for the production of natural rubber, which is essential for aviation and other industries. It is a perennial, monoecious outcrossing tree species of the Euphorbiaceae family, with its center of origin in the Amazon forests. The species, which contains 36 chromosomes (2n = 36), is considered to be an amphidiploid that has stabilized its genome during the course of evolution [1]. Domestication of *H. brasiliensis* started in 1876 with Sir Henry Alexander Wickham transporting 70,000 rubber seeds from Brazil to the Royal Botanic Gardens, Kew, England. This collection, which came from a small area near Manaus on the banks of Rio Tapajos in the upper Amazon, represented a very small proportion of the gene pool available throughout the Amazon [2]. Only 2700 of these seeds germinated. Most of the seedlings were shipped to Asian countries: 1919 to Sri Lanka (of which 90% survived), 18 to Indonesia (of which two survived) and 50 to Singapore (none of which survived). Subsequently, 22 additional plants were sent to Singapore, all of which survived. The surviving plants from these shipments formed the basis of the rubber plantation industry in the East today [3,4]. This narrow genetic base was further constricted by the use of a limited number of high-yielding clones for propagation and by the use of cyclical generation-wise assortative mating and selection for yield in breeding [5].

High-yielding *Hevea* clones are highly susceptible to biotic and abiotic stresses, which can significantly affect latex production. Biotic stress is mainly due to three major pathogens: *Phytophthora* spp., which causes shoot rot, abnormal leaf fall, patch canker, and black stripe diseases; *Corynespora cassiicola*, which causes *Corynespora* leaf fall disease; *Colletotrichum* spp., which causes *Colletotrichum* leaf fall disease [6,7] (Appendix A). Abiotic factors such as unfavorable climatic conditions, which adversely affect the growth and yield of rubber plants, act as predisposing factors for these diseases. Leaf fall caused by these diseases directly reduces plant growth and latex yield. Moderate to severe crop losses due to these diseases occur in all rubber-growing countries. Crop losses due to abnormal leaf fall disease in India, *Colletotrichum* leaf disease in Africa, and *Corynespora* leaf fall disease in China have been reported to range from 38–56% [8], from 7–45% [9], and from 20–25% [10], respectively. At present, the most extensively used management strategy for controlling these diseases is through the recurrent use of agrochemicals. While disease control achieved using genetics might provide sustainable intensification of crop production, introgression of disease resistance genes using backcrossing and phenotypic selection would be difficult for rubber because of its highly heterozygous nature, long breeding cycle, and the large area of land required for evaluation at each stage. Genetic mapping of resistance loci could enable the discovery of molecular markers that can be used to select resistant seedlings in breeding nurseries. 

The first linkage map of *Hevea* was constructed for an interspecific cross between *H. brasiliensis* and *H. benthamiana,* using restriction fragment length polymorphism (RFLP) and amplified fragment length polymorphism (AFLP) markers [11]. Subsequently, simple sequence repeat (SSR) markers were mapped for several cross combinations [12,13,14,15,16]. Recently, genotyping-by-sequencing (GBS) and transcriptome sequencing have been used to construct high-density linkage maps for intraspecific crosses of *H. brasiliensis* [17,18]. Intraspecific crosses of *H. brasiliensis* have been used to map quantitative trait loci (QTLs) for growth-related traits [15,19] and for resistance against South American leaf blight [13]. No such mapping has been conducted for resistance against the other major diseases of rubber. 

In the research that is presented here, we investigated genetic diversity and the genetic control of resistance to foliar diseases by applying DArTSeq genotyping-by-sequencing to a panel of *H. brasiliensis* clones and to progeny from an interspecific (*H. brasiliensis* × *H. benthamiana*) cross and conducted phylogenetic analysis and integrated genetic linkage mapping. 

## 2. Results

### 2.1. Genotyping-by-Sequencing, Phylogenetic Analysis, and Linkage Mapping

DArTSeq genotyping-by-sequencing of a 116-member diversity panel (Appendix A) and a set of 86 interspecific progeny (derived from crosses between *H. brasiliensis* clone RRII 105 and *H. benthamiana* clone F4542) generated 14,315 single nucleotide polymorphisms (SNPs) and 34,000 presence-absence SilicoDArT markers (Appendix A). The 116-member diversity panel consisted of 110 *H. brasiliensis* clones, one each of five other *Hevea* species (*H. benthamiana* F4542, *H. pauciflora*, *H. nitida*, *H. spruceana*, *H. camargoana*) and an interspecific hybrid FX 516 (*H. benthamiana* x *H. brasiliensis* AVROS 363). 

For the diversity panel, phylogenetic analysis was performed using data for 12,078 SNPs. Potential scale reduction factor values obtained using MrBayes Bayesian phylogenetic trees were between 1.0 and 1.1, and other examined parameters were within acceptable ranges. All *H. brasiliensis* clones were differentiated from individual clones of each of five other species (*H. benthamiana*, *H. camargoana*, *H. nitida*, *H. pauciflora*, *H. spruceana*) and from FX 516, an *H. benthamiana* × *H. brasiliensis* hybrid. Within *H. brasiliensis*, there were two main clades, one consisting of clones from Sri Lanka, Indonesia, and India, and the other consisting of clones from Malaysia and China. The clones from Malaysia showed the greatest diversity. The three clones from Indonesia were more similar to the clones from India than to the clones from Sri Lanka (Figure 1). 

Integrated linkage mapping performed with the software Lep-Map3 [20], using data for 23,978 markers (10,323 SNPs and 13,655 SilicoDArTs), resulted in a 3948 cM linkage map with 4757 loci across 18 linkage groups (Figure 2) (Appendix A). Of the 23,978 mapped markers, 17,310 (72.2%) were successfully anchored to positions on pseudomolecules in the GT1 reference genome assembly for *H. brasiliensis* [21], and a further 451 (1.9%) were anchored to unplaced GT1 scaffolds (Appendix A). Among the remaining markers, 3532 were anchored to positions in an earlier draft assembly [22], bringing the total percentage of anchored markers to 88.8%. 

A comparison of marker positions between the linkage map and the GT1 reference genome assembly indicated very high synteny and collinearity (Figure 3 and Figure 4, Appendix A). For some chromosomes (e.g., LG5 in Figure 4), the relationship between genetic distances was highly linear. For some other chromosomes, this relationship was non-linear, indicating a non-uniform distribution of recombination along chromosomes. For example, the terminal regions of the chromosome LG7 are more recombinogenic than the central region of that chromosome, and the opposite is true for LG8 (Figure 4).

### 2.2. Disease Resistance

For *Phytophthora* spp., which infects rubber pods, petioles, and mature leaves, the *H. brasiliensis* parent RRII 105 was classified as susceptible (rated 4 on a 5-point scale), and the *H. benthamiana* parent F4542 was classified as resistant (rated 2). Among 85 RRII 105 × F4542 F_1_ progeny assessed for resistance against *Phytophthora*, seven (8.2%) were highly resistant (rated 1), 22 (25.9%) were resistant (rated 2), 17 (20%) were moderately resistant (rated 3), 28 (32.9%) were susceptible (rated 4), and 11 (12.9%) were highly susceptible (rated 5) (Appendix A). 

For *Corynespora cassiicola* and *Colletotrichum acutatum,* which infect rubber tree leaves in the tender copper brown and light green stages, the *H. brasiliensis* parent RRII 105 was classified as highly susceptible (rated 5: leaf wilting always observed within 24 h of incubation with toxin extracts) and the *H. benthamiana* parent F4542 was classified as highly resistant (rated 1: no leaf wilting after 24 h of incubation with toxin extracts). Among 79 progeny tested with toxin extracts from *Corynespora cassiicola* and *Colletotrichum acutatum* extract, 43 (56.6%) were assigned intermediate ratings (2, 3, or 4) for both pathogens (Table 1).

Based on marker-trait association analysis, 12 SNPs were found to be significantly (*p* < 1.0 × 10^−4^) associated with one or more disease traits (Figure 5; Table 2): five (one on LG6 and four on LG18) for *Phytophthora* resistance, three (one on each of LGs 2, 4, and 18) for *Corynespora* resistance, and four (one on each of LGs 13, 14, and 17) for *Colletotrichum* resistance. In all cases, the favorable (resistance-associated) allele was from the *H. benthamiana* parent F4542. Of the 12 SNPs, 10 were anchored to the GT1 reference genome, and the other two were anchored to the rubber draft genome sequences. Except for one of the 12 markers, all were heterozygous in *H*. *benthamiana*. 

For the results of this work to be applied in marker-assisted rubber breeding, it could be useful to have simple assays for trait-associated SNPs. We designed Kompetitive Allele-Specific PCR (KASP) assays [23] for all 12 SNPs that showed significant associations with the disease traits (Appendix A). Those assays were then applied to the 178 RRII 105 × F4542 F_1_ progeny, and genotypes were clearly separated from each other (Figure 6; Appendix A). We looked at the two trait-associated SNPs on LG6 further, and for each of the SNPs, two genotypes (one heterozygous and one homozygous) had been observed in the DArTSeq results. With each of the KASP assays (WriKH1 for 100057258|F|0–6:T > C and WriHK7 for 100061575|F|0–46:G > A), both genotypes were observed and were clearly separated from each other (Figure 6).

For all 12 SNPs, when we compared the genotypic scores obtained from the DArTSeq with the KASP assays, the discrepancies varied from two to three: two for the marker 100061575|F|0–46:G > A (WriHK1) and three for the marker 100057258|F|0–6:T > C (WriKH7) (Appendix A). For four of these, DArTSeq calls were homozygous, and KASP calls were heterozygous. For the other 10 SNPs, there were no discrepancies between the genotypic scores obtained from the DArTSeq and the KASP assays.

Of the 178 F_1_ progeny studied, four of the F1 plants (9_26, 9_39, 9_76, and 13_1_34) carried the favorable alleles of *H. benthamiana* for all three disease traits. In addition, five of the F1 plants carried the favorable *H. benthamiana* alleles for resistance to *Phytophthora,* 22 carried the favorable alleles for *Corynespora*, and seven carried the favorable alleles for *Colletotrichum* (Appendix A). 

Of the four progeny with favorable alleles at all 12 SNPs, three had been included in the subset that was phenotyped for all three disease resistance traits. All three were rated as highly resistant to *Phytophthora* (rated 1 with a lesion size < 0.69 cm) and highly resistant and/or resistant to *Corynespora* and *Colletotrichum* (rated 1 and/or 2) (Appendix A). 

### 2.3. Significantly Associated Genes

The number of predicted genes near the trait-associated SNPs ranged from 67 to 36 (Appendix A). In the *Phytophthora*-associated region between 12 and 18.3 Mb on LG18, 55 genes were predicted with high confidence, including 12 with annotations related to innate immune response, host-pathogen relationships, or plant defense. Among these, XP_021650208.1 (disease resistance protein RPS2), GH714_038730 (BTB/POZ domain and ankyrin repeat-containing protein NPR1-like), XP_021650172.1 (heat shock factor protein HSF30-like), XP_021638503.1 (aquaporin NIP5-1), XP_021689264.1 (transcription factor BHLH089-like), XP_021640619.1 (zinc finger protein 6), XP_021665532.1 (cinnamoyl-CoA reductase 1) and GH714_038687 (WD repeat-containing protein VIP3) are similar to genes that have widely been reported to be associated with plant disease resistance and susceptibility. In the *Corynespora*-associated regions, between 44 and 45 Mb on LG18 and between 36.5 and 37.5 Mb on LG6, 67 and 26 genes were predicted, respectively. Among these, XP_021666909.1 (pathogenesis-related protein 5), XP_021647428.1 (WAT1-related protein At1g44800), XP_021660750.1 (ethylene-responsive transcription factor ERF014), and XP_021642270.1 (LRR receptor-like serine/threonine-protein kinase) are similar to genes that have been reported to be important in disease resistance in other plant species.

In the *Colletotrichum*-associated region between 6 and 6.7 Mb on LG14, 36 genes were predicted. Among these four had annotations associated with innate immune response (calcineurin B-like protein 7), biotic stress (O-acyltransferase WSD1-like), DNA methylation (DNA methylation 4-like), or systemic acquired resistance (laccase gene).

## 3. Discussion

In the research reported here, DArTSeq genotyping-by-sequencing was applied to a diversity panel and to a set of F_1_ progeny from an interspecific *Hevea* cross. Numerous SNPs and presence-absence polymorphisms were discovered in both sets of materials.

For the diversity panel, the DArTSeq genotypic data were used for phylogenetic analysis. In the resulting phylogenetic tree, *H. brasiliensis* clones clustered according to the country from which they were obtained, despite all cultivated rubber having been derived from a narrow genetic base. The clear differentiation among clones from Sri Lanka, Indonesia, India, Malaysia, and China may involve founder effects (related to the particular seedlings dispersed from England in the 19th century and/or the effects of subsequent breeding efforts in individual countries. It may also indicate that there has been limited international germplasm exchange in rubber breeding. Despite Malaysia not having been the recipient of any of the seeds originally distributed from England, the clones obtained from Malaysia exhibited the greatest diversity, which may reflect the use of parents from multiple sources. Both the number of markers and the number of clones investigated here were much larger than in a previous phylogenetic analysis based on the application of 30 EST-SSR markers to 51 clones [24]. That analysis also showed a separation based on country of origin, with most clones from Malaysia readily differentiated from and less diverse than clones from South America.

The parents of the interspecific cross, the *H. brasiliensis* clone RRII 105 and the *H. benthamiana* clone F4542, were chosen based on a need to improve resistance against foliar diseases that are important in India, combined with an interest in understanding the genetic basis of this resistance. RRII 105 is the most widely cultivated clone in the traditional rubber-growing regions of India. It is high yielding but highly susceptible to *Corynespora cassiicola* and *Colletotrichum acutatum* and moderately susceptible to *Phytophthora meadii*. In contrast, F4542 is low-yielding but highly resistant to all three of these fungal pathogens. In Brazil, F4542 has been used as a parent in breeding for resistance to *Phytophthora* spp. and *Microcyclus* [25]. 

The integrated map constructed here is by far the most saturated linkage map available for rubber, with between eight and ten times more markers than prior maps [11,12,13,14,15,16,17,18]. This is due to a combination of factors: the use of an interspecific cross, the use of GBS to discover sequence polymorphisms, and the use of Lep-Map 3 software [20], which can generate robust maps from low-coverage datasets. The integrated map generated here exhibited very strong synteny and collinearity with a recently generated reference genome for *H. brasiliensis* [20], providing a solid basis for future forward and reverse approaches to identify causal genes.

KASP assays designed from the DArTSeq tags confirmed, with very few exceptions, the polymorphisms discovered by DArTSeq (Appendix A). In four of five cases where marker calls differed, KASP calls were heterozygous, and DArTSeq calls were homozygous. This probably indicates that just one of the alleles was sequenced in sufficient depth by DArTSeq genotyping. This has previously been reported in switchgrass [26] and almond [27]. 

This is the first report of significant marker-trait associations for resistance against foliar diseases caused by *Phytophthora* spp., *Corynespora cassiicola,* and *Colletotrichum acutatum* in rubber. Similar to what has been reported for resistance to several other fungal pathogens of rubber [28,29,30], continuous variation was observed, and multiple loci of small effect were detected. Among the mapping progeny, four clones carried favorable alleles at all 12 resistance-associated SNPs. Breeding for resistance could begin with the crossing of selected interspecific progeny with high-yielding clones of *H. brasiliensis* (to introgress resistance into productive backgrounds) and/or with intercrossing among selected interspecific progeny (to pyramid resistance alleles at multiple loci). In either case, the DArTSeq genotypes generated here can be used to select parents and design crosses, while molecular marker assays, such as the KASP assays developed here, could be used in early-stage selection for resistance.

Anchoring significantly associated marker sequences to the 6, 14, and 18 pseudomolecules of GT1 rubber reference genome sequence assembly enabled the identification of predicted genes near the trait-associated SNPs (Appendix A). Given that there is very little known about resistance mechanisms and host-pathogen relationships of rubber foliage diseases, none of these predicted genes can be excluded based on annotation. However, some of the genes detected in the candidate regions belong to gene families that have been reported to be associated with disease resistance, host-pathogen relationships, and innate immune responses in other plant species (Appendix A). Therefore, this study provides new resources for candidate genes for predicting resistance in rubber foliage diseases that might lead to improvement in the speed of breeding for multi-genic traits and elucidate the molecular mechanisms to combat the devastating foliage diseases in rubber. 

## 4. Materials and Methods

### 4.1. Diversity Panel and H. brasiliensis (RRII 105) × H. benthamiana (F4542) Mapping Population

This research used a diversity panel consisting of 116 rubber clones (Appendix A) that belong to *H. brasiliensis* (54 from Malaysia, 51 from India, 5 from Sri Lanka, four from Indonesia, and 2 from China), each of one sample from *H. benthamiana*, *H. camargoana*, *H. pauciflora*, *H. spruceana,* and *H. nitida* and one sample from an interspecific hybrid between *H. benthamiana* and *H. brasiliensis* and a mapping population generated using an interspecific cross involving *H. brasiliensis* (RRII 105) and *H. benthamiana* (F4542) as the maternal and paternal parents, respectively. RRII 105 is the most popular clone in the traditional rubber-growing regions of India. It has a high yield but is moderately susceptible to *Phytophthora* spp. and highly susceptible to *Corynespora cassiicola* and *Colletotrichum* spp. F4542 has a low yield and is resistant to *Phytophthora* spp., *Corynespora cassiicola,* and *Colletotrichum* spp. The initial population was obtained in 2009 by performing hand pollination at the Rubber Research Institute of India (RRII). Due to the asynchronous flowering of the parents, pollination success was poor, and few progeny were obtained. Crossing was therefore repeated from 2011 to 2015 until sufficient F_1_ progeny were obtained. The final mapping population consisted of 178 F_1_ progeny, all of which are maintained in field nurseries using standard management practices. 

### 4.2. DNA Extraction and DArTSeq Genotyping-By-Sequencing 

Genomic DNA was extracted from 500 mg of lyophilized tender green leaf tissue following the CTAB method [31]. DNA quality was tested by electrophoresis on an ethidium bromide-stained 1% agarose gel, and quantity was measured using a NanoDrop™ spectrophotometer. Non-hybrids were eliminated based on the results of SSR marker *hmCT44* (GenBank Acc. No. AY962210) (Appendix A). PCR reaction was carried out in a 10 µL final volume containing 20 ng of genomic DNA, 0.2 µM each of the forward and reverse primers (forward: 5′ TCTCATCCATGCAAGAACCCTA 3′ and reverse: 5′ GCGTTCCCAAATGCATACCT 3′), 200 μM dNTPs and 0.4 U of *Taq* DNA polymerase (GE Healthcare, UK). The thermocycling conditions were initial denaturation for 5 min at 95 °C followed by a touch-down PCR program for 7 cycles of 94 °C for 30 s, 63 °C for 1 min, Δ↓ 1 °C for 7 cycles, and 72 °C for 1 min. This was followed by normal cycling of 94 °C for 30 s, 56 °C for 1 min, 72 °C for 1 min for 23 cycles, and a final extension at 72 °C for 10 min. Once the PCR was completed, reactions were stopped immediately by the addition of 10 µL formamide loading buffer, and the amplification products were run on a 6% denaturing polyacrylamide gel containing 7 M urea using 0.6× TBE buffer at a constant power of 55 W. Gels were silver stained following the protocol described by Roy et al. [32] and samples that showed to be non-hybrids were removed from further analysis. The DArTSeq genotyping-by-sequencing technique (www.diversityarrays.com/dart-application-dartseq, accessed on 22 November 2022) was applied to DNA aliquots of parents and a randomly selected subset of 86 progeny at Diversity Arrays Technology (Bruce, ACT, Australia). In brief, DNA samples were digested with a combination of *Pst* I/*Mse* I restriction enzymes, and then the multiplexed reduced representation library was sequenced using single-end sequencing on an Illumina HiSeq 2500 with running 77 cycles. Data were provided by Diversity Arrays Technology, Australia, with polymorphisms scored as either co-dominant single nucleotide polymorphisms (SNPs) or as dominant (presence/absence) SilicoDArT sequence tags. 

### 4.3. Phylogenetic Analysis

Phylogenetic analysis was performed using the DArTSeq data generated from the *Hevea* diversity panel of 116 rubber clones using MrBayes v3.2.6 [33] with the following parameters: a general time reversible model and a gamma-shaped distribution of rates across sites function were used. The Markov Chain Monte Carlo (MCMC) was set to three million generations with a sampling frequency of 100 and 250 burn-in. The analysis used all DArTSeq data from a VCF file with less than 5% missing data and for which the minor allele frequency was at least 5%. For the tree construction, diagnostic parameters such as the Potential Scale Reduction Factor (PSRF), shape of gamma distribution rate variation, and stationary rate frequency were considered to determine if the obtained tree is optimal and trustable. Chain convergence was checked in Tracer version 1.6 [34] by examining the log-likelihood plots, and the effective sample size values were ensured to be well above 200. The resulting phylogenetic tree was visualized with FigTree v.1.4.4 [31]. 

### 4.4. Construction of Integrated Genetic Map Using Lep-Map3 Software

For the construction of the integrated genetic map, all SNP markers that did not deviate from expected segregation patterns (1:1 or 1:2:1) and all SilicoDArT markers with less than 20% missing data points were selected. VCF tools [35] were used for filtering the VCF file, and SilicoDArT markers were coded as homozygote indels based on the parent, as mentioned in Lep-Map3. An integrated map was generated following the instructions in Lep-Map3 v0.2 [20] software, together with a pedigree file indicating the parents of the controlled cross. The modules in Lep-Map3 included several steps starting from ParentCall2 to remove erroneous or missing parental genotypes; Filtering2 to remove markers with high segregation distortion (*p*-value < 1 × 10^−3^); SeparateChromosome2 to assign markers into linkage groups by computing all pair-wise LOD scores between markers and to join markers with a user define LOD threshold; JoinSingles2All to assign singular markers to existing LGs by computing LOD scores between each single marker and markers from the existing LGs and OrderMarkers2 to order the markers within each LG by maximizing the likelihood of the data given in the order. To order markers within the linkage group, a LOD score of 5 was used. A total of iteration 100 was used to obtain a final map. 

### 4.5. Genome Scaffold Anchoring and Comparative Mapping between the Integrated Map and the Rubber Reference Genome

All unique DArTSeq sequence reads mapped in the integrated map that was at least 64 bp long were aligned with the rubber reference genome sequence [36] using the BLAST+ tool version 2.12.0 (http://www.ncbi.nlm.nih.gov/blast, accessed on 22 November 2022). Each sequence read was considered to have been anchored to the rubber reference genome if it mapped to a unique site with >90% sequence similarity and an E-value < 1 × 10^−15^. Sequences that met these criteria were selected to compare the genetic positions in the integrated map with physical positions in 18 main pseudomolecules of the rubber reference genome. A circular plot was drawn using the Circlize R package version 0.4.15 [37]. 

### 4.6. Phenotypic Trait Measurement for Disease Resistance

Disease resistance was assessed for three major pathogens of rubber: *Phytophthora meadii*, *Corynespora cassiicola,* and *Colletotrichum acutatum* through in vitro studies. The parents (RRII 105 and F4542) and 86 F_1_ progeny were phenotyped. To maintain uniformity in the leaf growth stage during inoculation processes, phenotyping was carried out using the initial population that was crossed in 2009. Due to the non-availability of the physiologically same stage of leaves with some progeny, *Phytophthora* resistance was assessed with 85 progeny, and resistance for *Corynespora* and *Colletotrichum* was assessed with 79 progeny. 

#### 4.6.1. Screening for Phytophthora Resistance Using Zoospore Suspension

The plants were cut back for uniform growth and maturity. Mature leaves (two months old) were collected from each progeny and transported to the laboratory with the petiole dipped in water. Leaf discs of 3.5 cm diameter were taken from the leaves using a punching device. Four discs per leaf were taken, with a total of 32 discs per progeny. Leaf discs were placed with their abaxial surface upwards in Petri plates previously lined with three sterilized moist filter papers and kept in the inoculation room at 25 °C. The center of leaf discs was inoculated with 20 µL drops of spore suspension containing 2 × 10^5^ zoospores/mL [38,39]. Following inoculation, the Petri plates were incubated at a temperature of 24 °C under alternate light and dark conditions. Leaf discs of RRIM 600, a highly susceptible clone, and FX 516, a tolerant clone to abnormal leaf fall (ALF) disease, were maintained as control. Disease severity was assessed by measuring the size of lesions developed on the leaf discs periodically from 72–144 h after inoculation. For *Phytophthora* resistance, the greatest differentiation was found at 96 h inoculation, and severity scores obtained at 96 h were therefore used for subsequent analysis. Screening was repeated three times, and the progeny were classified into five categories based on the necrotic area/lesion size: highly resistant (0.0–0.69 cm); resistant (0.7–1.39 cm); moderately resistant (1.4–2.09 cm); susceptible (2.1–2.79 cm), and highly susceptible (2.8–3.5 cm) (Appendix A). 

#### 4.6.2. Toxin-Based Screening for *Corynespora* and *Colletotrichum* Resistance

In order to screen for disease resistance of *Corynespora cassiicola* and *Colletotrichum acutatum*, toxin-based screening was employed using toxins extracted from the respective pathogen. Pure single spore cultures of *Corynespora cassiicola* and *Colletotrichum acutatum* were grown on potato dextrose agar medium. Twelve discs of 0.8 cm size from 10-day-old cultures were transferred aseptically to 100 mL of modified Czapek Dox liquid medium [40,41] and incubated without agitation for 12–14 days in laboratory conditions (25 ± 2 °C). The culture filtrate was extracted using a vacuum filtration unit, and the crude toxin was used for screening. Glass vials of 15 mL capacity were used with 5 mL of crude toxin in a dilution of 1:6 in modified Czapek Dox liquid medium diluted in water. Healthy leaves collected in the morphogenetic stage C (limp, brownish to light green) [42] were excised underwater. The petioles of excised leaflets were immediately transferred to vials containing the diluted toxin. For each of the screening, similar stage leaves from each of the parents (*H. brasiliensis* is highly susceptible and *H. benthamiana* is highly tolerant) were used as controls in modified Czapek Dox liquid medium without crude toxin from pathogens. The leaflets were observed regularly for any signs of drooping [43]. Wilting intensity (water loss estimation) was visually assessed at 24 and 48 h following the treatment. Scoring for disease resistance was made on a scale of 1–5, with 1 indicating high resistance (with leaves remaining fresh even after 48 h of incubation in the toxin) and 5 indicating high susceptibility (with leaves completely wilted after just 24 h of incubation in the toxin) (Appendix A). Ten replicates for each progeny were evaluated, and the experiment was repeated three times. 

### 4.7. Marker Trait Association Analysis Using Integrated Map

Markers associated with each of these disease resistance traits were identified by performing association mapping analysis using TASSEL V5.2.73 software [44] using the Q-model (GLM Q-matrix as correction for population structure) implemented in General Linear Model (GLM) method. Marker alleles with *p* values 1 × 10^−4^ were declared significantly associated with each of the disease resistance, and a standard Bonferroni procedure was applied at *p* < 0.000001 [45].

### 4.8. Development and Use of KASP Assays Where Significant Marker-Trait Association Were Detected

For the SNPs where significant marker-trait associations were detected, DArTSeq tag sequences were selected for the development of allele-specific assays. First, these sequences were aligned with *Hevea* draft genomic contig sequences using the Map to Reference function in Geneious software version 10.1.8 [46] to obtain sequences of at least 100 bp. Based on these sequences, primer sets (each consisting of two allele-specific primers and one common primer) were designed using Kraken™ software (LGC Ltd., Teddington, UK). The primer sets were named using the prefixes WriHK, with Wri referring to the Waite Research Institute, H for *Hevea*, and K referring to KASP (Appendix A). Each KASP assay was tested with 178 (the 86 progeny that were used for DArTSeq analysis plus 96 additional progeny) of the *H. brasiliensis* × *H. benthamiana* F_1_ progeny, with a no-template (water) sample included as a negative control. 

For KASP assays, 1.972 µL of 1× KASP Master Mix (LGC Ltd., Teddington, UK) was added to 10 ng of DNA (5 μL of 2 ng/μL dried at 55 °C for 1 h). An aliquot of a primer mixture (0.028 µL, containing 12 μM of allele-specific forward and reverse primers and 30 μM of common primer) was added to each sample. The thermal cycling conditions for fragment amplification comprised two temperature steps in a Hydrocycler-16 PCR system (LGC Ltd., Teddington, UK). DNA was denatured at 94 °C for 15 min, followed by 10 cycles of 94 °C for 20 s, 61–55 °C for 60 s (dropping 0.6 °C per cycle), 26 cycles of 94 °C for 20 s, and 55 °C for 60 s. Fluorescence was detected using a Pherastar^®^ Plus plate reader (BMG LABTECH, Ortenberg, Germany). Three further cycles (94 °C for 20 s and 57 °C for 60 s) were carried out, with fluorescence detected after each cycle. Data from the cycle that yielded the best separation among genotypic clusters were analyzed using Kraken™ software (LGC Ltd., Teddington, UK).

### 4.9. Candidate QTL Regions and Potential Key Genes

For each of these disease traits, the close proximity regions to the SNPs that showed significant marker-trait associations and which aligned with the GT1 rubber genome sequences were defined as the candidate QTL regions. Within those regions, the genes that have been reported to be associated with disease resistance, plant-pathogen relationships, and immune responses were selected as potential key genes for rubber foliage disease resistance traits. 

## Figures and Tables

**Figure 1 plants-11-03418-f001:**
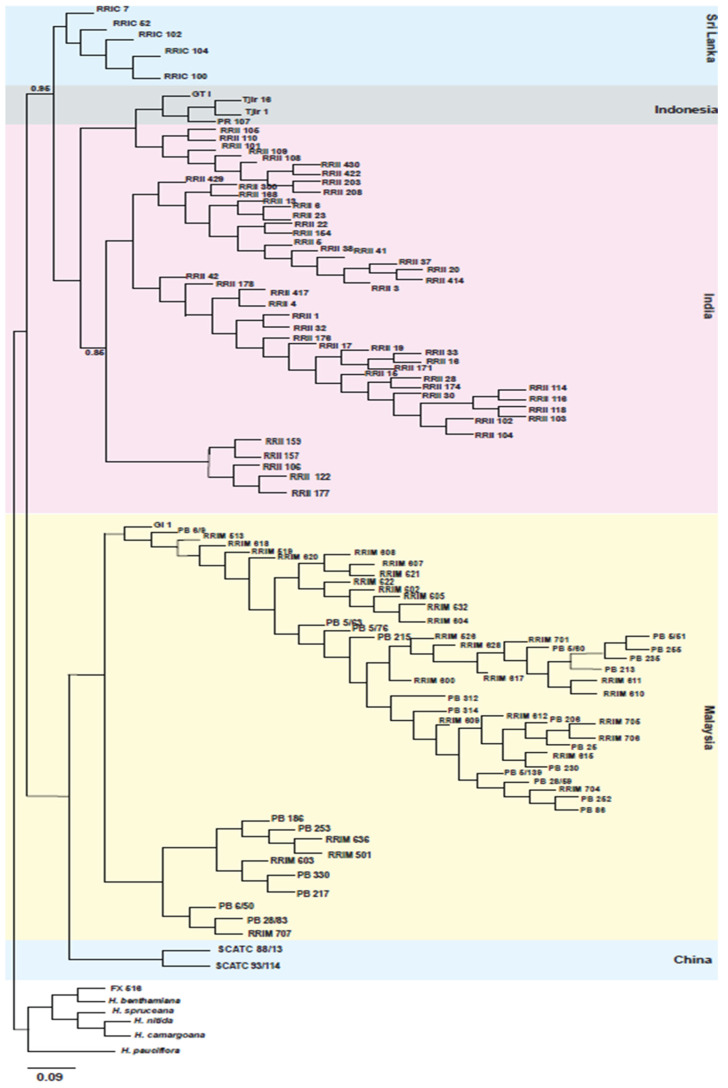
Phylogenetic tree constructed for 116 clones using SNP markers obtained from DArTSeq. Values indicated on the nodes are Bayesian posterior probabilities and are 1.0 unless otherwise indicated. The scale bar represents the probability of nucleotide substitutions per site.

**Figure 2 plants-11-03418-f002:**
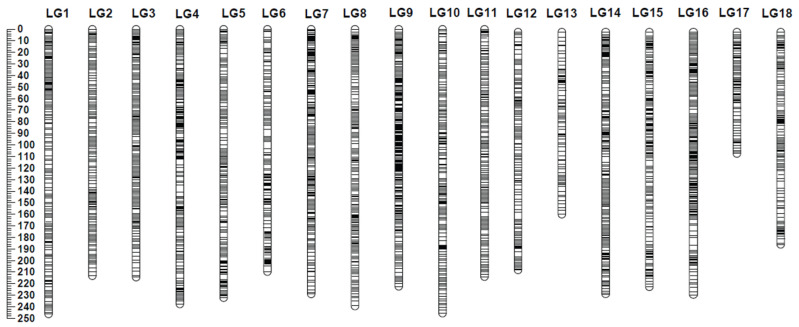
Positions at which markers were mapped on an 18-linkage-group integrated map constructed from genotyping-by-sequencing data for 178 RRII 105 × F4542 progeny. The scale to the left of the map shows genetic distances in cM.

**Figure 3 plants-11-03418-f003:**
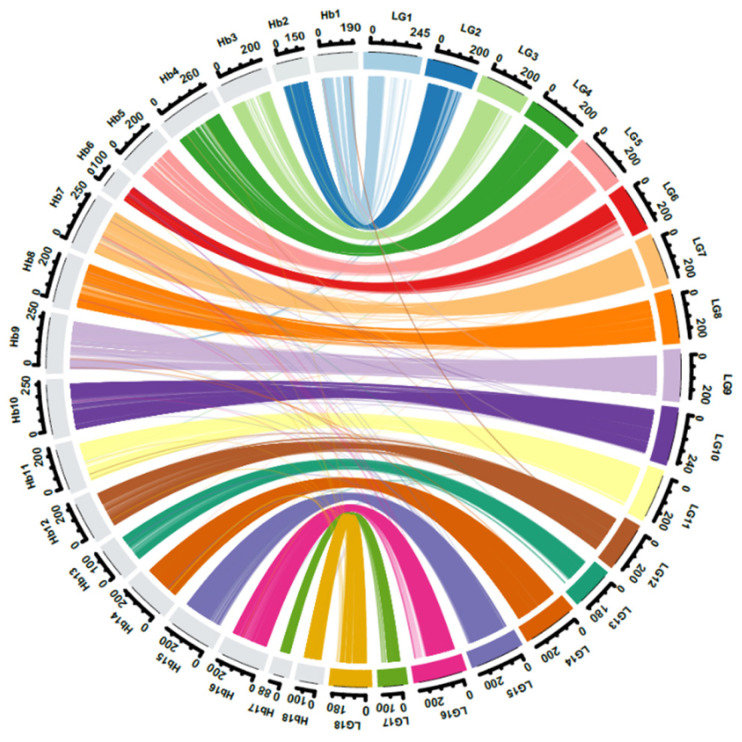
Synteny and collinearity of 17,310 markers between the RRII 105 × F4542 linkage map and the reference genome of *Hevea brasiliensis.* The outer circle indicates the genetic lengths of linkage groups LG1 through LG18 in cM and the physical lengths of pseudomolecules Hb1 through Hb18 in Mb.

**Figure 4 plants-11-03418-f004:**
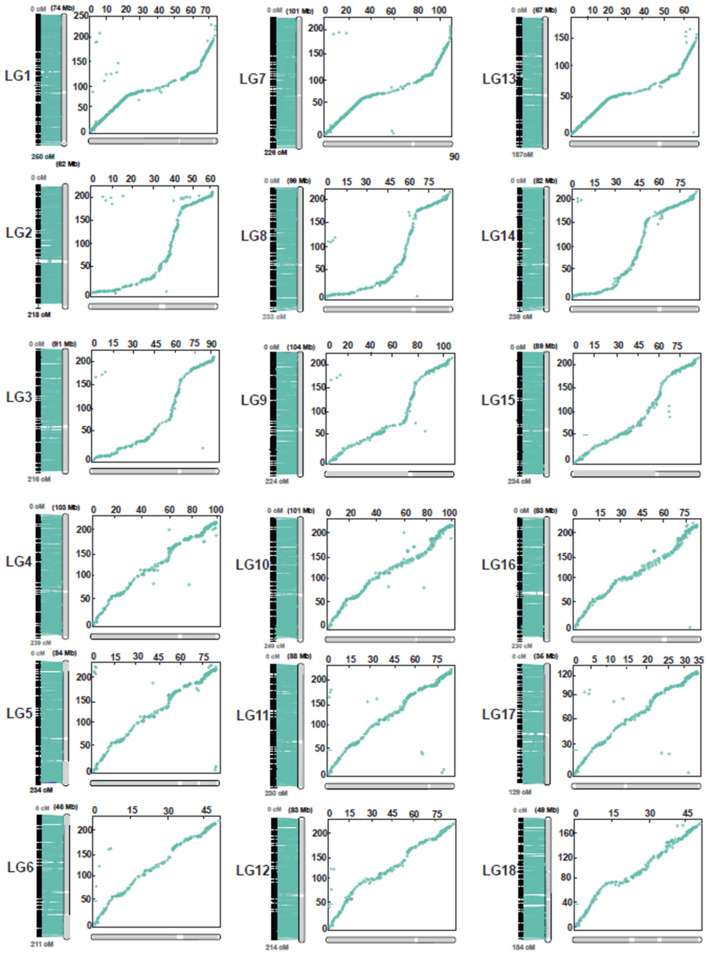
Relationship between the linkage map and the reference genome of *Hevea brasiliensis*. Each panel shows the genetic positions of markers on the linkage map (vertical axis, in cM) plotted against the physical positions of the same markers on the corresponding pseudomolecule of the reference genome (horizontal axis, in Mb). The right part represents the genetic and physical locations of the markers.

**Figure 5 plants-11-03418-f005:**
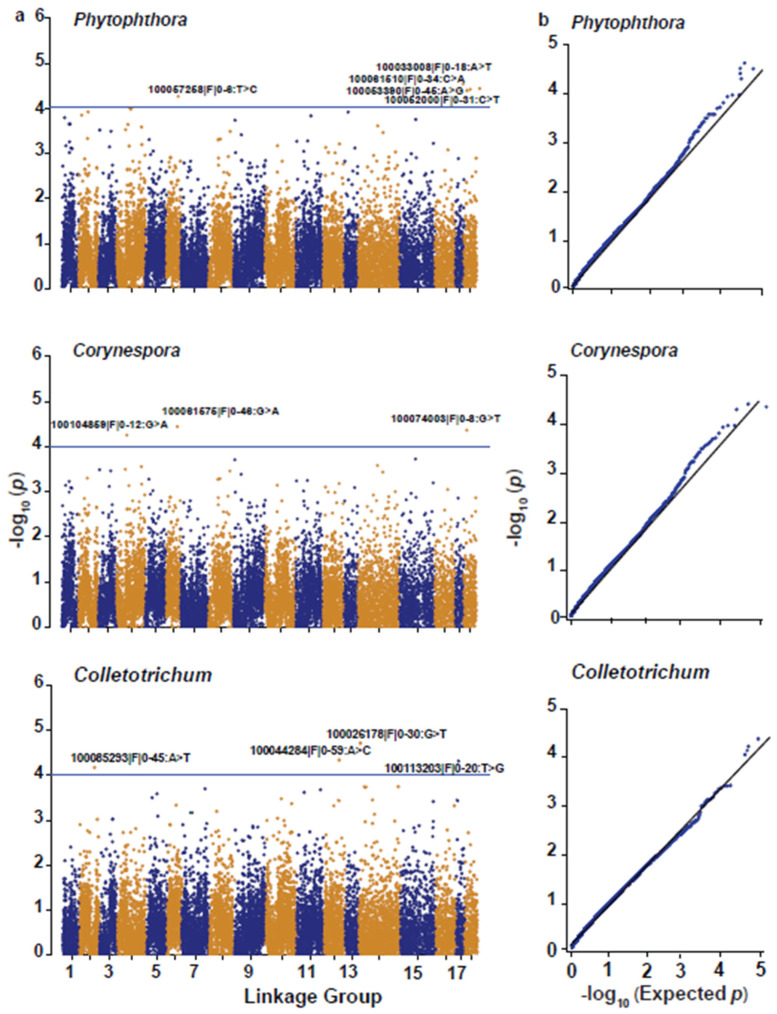
Manhattan plots and quantile-quantile plots from marker-trait association analysis for *Phytophthora*, *Corynespora,* and *Colletotrichum* resistance traits. In the Manhattan plots (**a**), each dot represents an SNP, showing the significance of its association with disease resistance (−log_10_
*p*-value) (vertical axis) plotted against its genetic position (horizontal axis), and the horizontal blue lines indicate the genome-wide significance threshold of *p* = 1.0 × 10^−4^. In the quantile-quantile plots (**b**), the black line represents the 95% confidence limit under the null hypothesis of no-marker-trait association, and the black dots represent *p*-values.

**Figure 6 plants-11-03418-f006:**
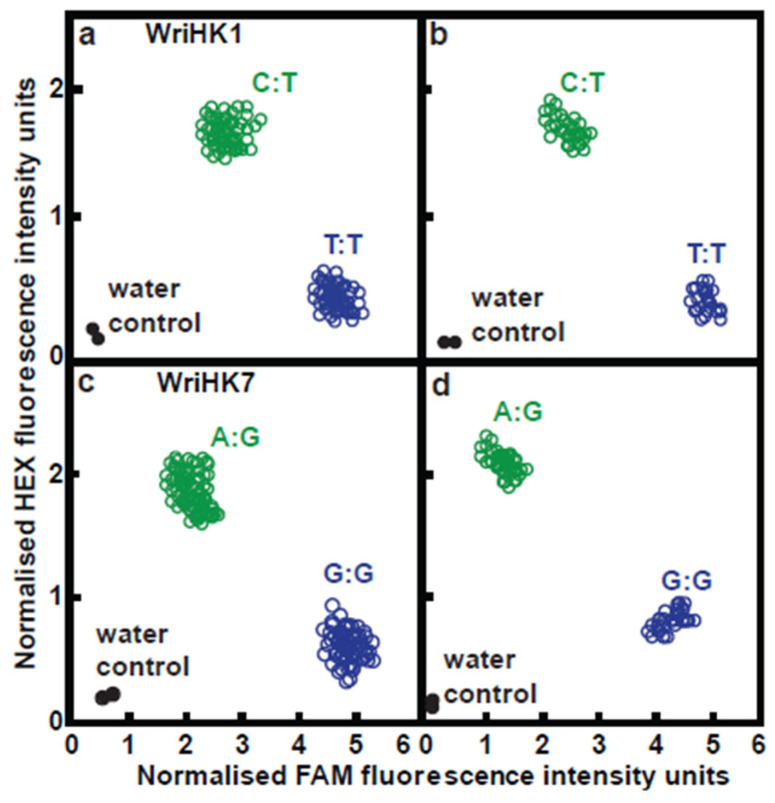
Results obtained with markers 100057258|F|0–6:T > C (WriKH1) and 100061575|F|0–46:G > A (WriHK7) on LG6. For both markers, left panels (**a**,**c**) show the marker results obtained for 178 *H. brasiliensis* × *H. benthamiana* progeny using KASP markers, and right panels (**b**,**d**) show the marker results obtained for 86 progeny using DArTSeq.

**Table 1 plants-11-03418-t001:** Numbers (and percentages) of *H. brasiliensis* x *H. benthamiana* F_1_ progeny classified between 1 (highly resistant) and 5 (highly susceptible) for resistance against *Colletotrichum acutatum* and *Corynespora cassiicola*.

*Colletotrichum acutatum* Resistance Classification	*Corynespora cassiicola* Resistance Classification					Total
	1	2	3	4	5	
**1**	2	2	2	1	2	9 (11.3%)
**2**	4	8	5	6	3	26 (32.9%)
**3**	1	9	4	6	5	25 (31.6%)
**4**	1	5	3	4	1	14 (17.8%)
**5**	0	1	2	1	1	5 (6.4%)
**Total**	8 (10.1%)	25 (31.6%)	16 (20.2%)	18 (22.8%)	12 (15.2%)	79 (100%)

**Table 2 plants-11-03418-t002:** Markers that showed significant associations with disease traits in the integrated map of *Hevea brasiliensis* and *H. benthamiana*.

Pathogen Associated with the Disease Trait	Linkage Group	Marker	Reference Genome Sequence	Physical Position (bp)		−Log_10_ (*p*)	R^2^ (%) (Variance Explained)	Allele Effect
				Start	End			
*Phytophthora*	LG6	100057258|F|0–6:T > C	KB619684.1_scaffold 218036	28,834	28,903	4.28	14	1.10
	LG18	100033008|F|0–18:A > T	GT1 Reference genome (CM021243.1)	18,128,717	18,128,649	4.52	18	3.21
		100061510|F|0–34:C > A	GT1 Reference genome (CM021243.1)	12,918,745	12,918,685	4.25	14	2.57
		100053390|F|0–45:A > G	GT1 Reference genome (CM021243.1)	15,754,360	15,754,413	4.25	14	2.57
		100052000|F|0–31:C > T	GT1 Reference genome (CM021243.1)	15,953,978	15,953,939	4.21	14	2.32
*Corynespora*	LG4	100104859|F|0–12:G > A	GT1 Reference genome (CM021229.1)	34,907,019	34,906,974	4.21	14	2.30
	LG6	100061575|F|0–46:G > A	GT1 Reference genome (CM021231.1)	37,466,880	37,466,948	4.35	16	2.43
	LG18	100074003|F|0–8:G > T	GT1 Reference genome (CM021243.1)	44,256,460	44,256,528	4.35	16	2.41
*Colletotrichum*	LG2	100085293|F|0–45:A > T	AJJZ010325919.1_contig 401252	1511	1580	4.13	11	1.25
	LG12	100044284|F|0–59:A > C	GT1 Reference genome (CM021237.1)	73,830,891	73,830,823	4.26	14	1.56
	LG14	100026178|F|0–30:G > T	GT1 Reference genome (CM021239.1)	6,682,648	6,682,714	4.48	16	0.98
	LG17	100113203|F|0–20:T > G	GT1 Reference genome (CM021242.1)	2,385,929	2,385,861	4.21	16	0.97

## Data Availability

The datasets generated and analysed during this study are available in the article and in its online supplementary materials.

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
