# Peer review of "Analysis of Genetic Diversity and Resistance to Foliar Pathogens Based on Genotyping-by-Sequencing of a Para Rubber Diversity Panel and Progeny of an Interspecific Cross"

_plants, 2022, doi:10.3390/plants11243418_

Round 1
Reviewer 1 Report
Dear authors and the editor,
This study focus on the genetic diversity and resistance to foliar pathogens by-sequencing of para rubber tree, and an integrated linkage map was constructed which will be good for breeding and genetic research. But there are still some problems to be answered and improved.
I. Firstly, we could not see the profile of para rubber tree infected by phytophthora spp., corynespora cassiicola, or colletotrichum spp., in my opinion, which is very necessary for this study.
II. Althought there are some statments in the text, but the abstract lacks of a useful and concise description on analysis of genetic diversity of rubber genotypes, please consider it.
III. Please consider again the descriptions of the abstract and result of candidate QTL regions and potential key genes which are greatly for this study , but it seems no highlights.
IV. Please read and choose some new references related with genome and disease resistance of para rubber tree.
Reviewer 3 Report
No doubts that the economic importance of rubber tree justifies this research. The fact that rubber tree industry in Asia lies on a very limited genetic bases makes the study of genetic variability of rubber tree very intriguing and the results of the research presented in this paper very relevant. From a genetic improvement point of view, rubber tree is a very difficult case: Hevea is monoecious and outcrossing and has a very long life cycle. The results reported in this paper can offer good support in this respect. The research design appears scientifically robust, the results are convincing and clearly presented and support well the conclusions presented in the discussion section. The only doubt I have about the publication of this paper is that the research results would maybe deserve to be presented in two distinct papers (one on genetic diversity and one on genetic resistance to foliar pathogens). Supplementary figure 1 is superfluous, but, being supplementary, does not harm.
